# In Vitro Assessment of Biofilm Production, Antibacterial Resistance of *Staphylococcus aureus*, *Klebsiella pneumoniae*, *Pseudomonas aeruginosa*, and *Acinetobacter* spp. Obtained from Tonsillar Crypts of Healthy Adults

**DOI:** 10.3390/microorganisms11020258

**Published:** 2023-01-19

**Authors:** Renata Viksne, Karlis Racenis, Renars Broks, Arta Olga Balode, Ligija Kise, Juta Kroica

**Affiliations:** 1Department of Otorhinolaryngology, Daugavpils Regional Hospital, LV-5401 Daugavpils, Latvia; 2Department of Doctoral Studies, Riga Stradins University, LV-1007 Riga, Latvia; 3Department of Biology and Microbiology, Riga Stradins University, LV-1007 Riga, Latvia; 4Center of Nephrology, Pauls Stradins Clinical University Hospital, LV-1002 Riga, Latvia; 5Department of Microbiology, NMS Laboratory, LV-1039 Riga, Latvia

**Keywords:** biofilm, colonization, tonsillar microbiota, tonsillar crypts

## Abstract

Background and Objective: Tonsillar crypts can be considered a reservoir for a variety of bacterial species. Some bacterial species can be considered part of the normal oropharyngeal microbiota. The roles of other pathogens, for example, the so-called non-oral and respiratory pathogens *Staphylococcus aureus*, *Klebsiella*, *Pseudomonas*, and *Acinetobacter* spp., which have strong virulence factors, biofilm production capacity, and the ability to initiate infectious diseases, are unclear. The purpose of this study was to detect the presence of *S. aureus*, *K. pneumoniae*, *P. aeruginosa*, and *Acinetobacter* spp. within the tonsillar crypts of healthy individuals, and to analyze the pathogens’ biofilm production and antibacterial resistances. Results: Only common oropharyngeal microbiota were cultivated from 37 participant samples (40.7%). The most commonly isolated pathogenic bacterium was *S. aureus*, which was isolated in 41 (45%) participant samples. *K. pneumoniae* was isolated in seven (7.7%) samples, *Acinetobacter* spp. were isolated in five (5.5%) samples, and *P. aeruginosa* was isolated in two (2.2%) samples. Biofilm producers predominated among the pathogenic bacteria; 51 strains were biofilm producers, and among them, 31 strains were moderate or strong biofilm producers. The tested *S. aureus*, *K. pneumoniae*, *P. aeruginosa*, and *Acinetobacter* spp. strains were sensitive to commonly used antibiotics (amoxicillin–clavulanic acid, clindamycin, or ciprofloxacin). One of the isolated *S. aureus* strains was MRSA. Conclusions: Biofilm is a commonly observed feature that seems to be a naturally existing form of pathogenic bacteria colonizing human tissue. *S. aureus*, *K. pneumoniae*, *P. aeruginosa*, and *Acinetobacter* spp. occasionally occur in the tonsillar crypts of healthy individuals, and, therefore, it is most likely that *S. aureus, K. pneumoniae, P. aeruginosa*, and *Acinetobacter* spp. in opportunistic tonsillar infections originate from the tonsillar crypt microbiota.

## 1. Introduction

All mucosal surfaces of the human body are colonized by a plethora of bacterial communities [1]. Different oral structures and tissues are colonized by distinct microbial communities [2]. Oral microbiota have been shown to be functionally connected to infectious and inflammation-related diseases [3]. The palatine tonsils are mucosa-associated and immunocompetent lymphoid organs localized on the lateral wall of the oropharynx [3]. They are continuously exposed to bacteria from saliva, inhaled air, ingested food, and the airway surface liquids of the respiratory tract, and play an essential role in the human immune defense system via surveillance, detection, and the initiation of an immune response [4]. Both the surface of the tonsils and the extensive tubular tonsillar crypts are important colonization sites for many pathogenic and commensal microorganisms [5]. Tonsillar infections may stem from bacteria within the tonsillar crypts or the parenchyma, rather than from those on the surface [6]. The microbiota of the palatine tonsils play an important role in health through the etiology of infection and the carriage of adventitious pathogens.

The oropharyngeal microbiome has been extensively characterized through cultivation and culture-independent molecular methods [1,2,3]. Oropharyngeal bacterial communities are dominated by six major phyla, *Firmicutes*, *Bacteroidetes*, *Proteobacteria*, *Actinobacteria*, *Spirochetes*, and *Fusobacteria*, representing 96% of all taxa found in the oropharynx [1,2,7]. Less dominant taxa are highly specific to both individuals and body habitats [8]. In the oral cavity, most habitats are dominated by *Streptococcus*, and these are followed in abundance by *Haemophilus* in the buccal mucosa, *Actinomyces* in the supragingival plaque, and *Prevotella* in the subgingival plaque [8]. Less dominant taxa, species that pose a modest degree of risk, and various clinically important pathogens which are generally considered non-oral bacteria, such as Gram-negative enteric rods, enterococci, and staphylococci, are highly important [9]. Disease states are often associated with a disruption of the microbial community, frequently resulting in one or a few pathogenic organisms emerging. *Staphylococcus (S.) aureus* is of particular interest as the cause of methicillin-resistant *S. aureus* (MRSA) infections, as are the respiratory bacterial pathogens *Klebsiella (K.) pneumoniae*, *Pseudomonas (P.) aeruginosa*, and *Acinetobacter* spp., which are potent biofilm producers ([9,10,11], p. 2). The pathogens mentioned above are among the ESKAPE pathogen group (*Enterococcus faecium*, *Staphylococcus aureus*, *Klebsiella pneumoniae*, *Acinetobacter baumannii*, *Pseudomonas aeruginosa*, and *Enterobacter* species) and have been declared critical priority pathogens by the World Health Organization due to their increasing levels of resistance to commonly used antibiotics [12].

Knowledge regarding biofilms has significantly increased over the years, from the attachment of biofilms to artificial surfaces to mucosal biofilms. Studies have revealed that mucosal biofilms exist in both healthy and diseased individuals, and that the presence of a mucosal biofilm is not always associated with disease [13]. Biofilms may exist in the palatine tonsils of healthy adults due to cryptic tissue structure, a temperature lower than physiological body temperature, and direct, repeated exposure to respiratory bacterial pathogens [10]. Signs that differentiate between “healthy” and “pathological” biofilms are currently being sought [10].

Biofilms play a role in the process of chronic and recurrent infections due to certain important pathology-associated features of biofilms, including enhanced resistance to antibiotic treatments and increased host defense [14]. Biofilm-associated bacteria can be up to 1000 times more resistant to antimicrobial agents relative to planktonic bacteria [10,13]. Mucosal biofilm has been implicated in relation to recurrent tonsillitis [15].

Based on our hypothesis that tonsillar crypts are richly colonized and covered with bacterial biofilm even in the absence of disease, the aim of this study was to detect the presence of the so-called non-oral and respiratory pathogens *S. aureus*, *K. pneumoniae*, *P. aeruginosa*, and *Acinetobacter* spp. within the tonsillar crypts of healthy medical students, and to analyze the pathogens’ biofilm production and antibacterial resistance. Medical students are present in medical institutions more often than the general public, so there is a higher likelihood of them being carriers of pathogenic strains.

## 2. Materials and Methods

A total of 91 healthy students from the Medical Faculty of Riga Stradins University were included in a prospective cohort study from 1 October 2019 to 31 December 2019. The healthy individuals were fifth-semester Medical Faculty students for whom otolaryngology was the first clinical subject taught at the hospital, and they had not yet been exposed to patients. The inclusion criteria were absence of tonsillar pathologies or upper respiratory tract infections at the time of data collection, no comorbidities, and no antibacterial therapy for at least 4 weeks. Those who received antibacterial therapy in the last 4 weeks, had unhealthy oral conditions (including tooth decay and periodontal disease), had prosthetic devices within the oral cavity, or who failed to give consent were excluded.

The study protocol was approved by the local ethics committee of Riga Stradins University (document no. 49/30.11.2017), and all the data were collected according to the relevant guidelines on data protection and confidentiality. Written informed consent was obtained from all subjects before the study.

### 2.1. Sample Collection

Samples for microbiological testing were obtained from tonsillar crypts using a brush (Kito brush, reference number 0640, Kaltek srl, Padova, Italy).

### 2.2. Isolation of Microorganisms and Microbiological Investigation

Material from tonsillar crypts were taken, placed, and transported in AMIES transport medium at room temperature within 24 h and cultivated on two Columbia blood agar plates with and without optochin disk, Brucella blood agar, Chocolate agar with oleandomycin disc, Mannitol salt agar, MacConkey agar, and Sabouraud dextrose agar plates. Columbia blood agar, Mannitol salt agar, MacConkey agar, and Sabouraud dextrose agar plates were incubated at 36 ± 1 °C for 24–48 h aerobically. A Brucella blood agar plate was incubated in a BD GasPak™EZ pouch system at 36 ± 1 °C for up to five days. A Columbia blood agar plate with an optochin disc incubated in a CO_2_ incubator at 36 ± 1 °C for 24–48 h was used for the cultivation of *Streptococcus pneumoniae*. A Chocolate agar plate with an oleandomycin disc incubated in a CO_2_ incubator at 36 ± 1 °C for 24–48 h was used for the cultivation of *Haemophylus* spp. We took note of the common oropharyngeal microbiota as described by the European Society of Clinical Microbiology and Infectious Diseases [16]. Microorganisms that are not part of the common oropharyngeal microbiota were considered as potential pathogens. The identification of the considered pathogens was performed using a Microflex LT (Bruker Daltonics flex Analysis version 3.4, Bruker Daltonics GmbH & Co. KG, Bremen, Germany) matrix-assisted laser desorption ionization–time-of-flight mass spectrometer (MALDI–TOF MS) system.

### 2.3. Antibacterial Susceptibility Testing

Susceptibility testing was performed by the Kirby–Bauer disk diffusion method. Overnight cultures were suspended in physiological saline to 0.5 McFarland units (McFarland Densitometer DEN-1, Biosan, Latvia). The suspension was inoculated on Mueller–Hinton agar (Oxid, UK). Selected antibiotics were placed on the inoculated plates. For *S. aureus* strains, cefoxitin 30 µg, ceftriaxone 30 µg, benzylpenicillin 1iu, ampicillin 2 µg, ampicillin–sulbactam 10/10 µg, amoxicillin–clavulanic acid 20/10 µg, norfloxacin 10 µg, amikacin 30 µg, erythromycin 15 µg, clindamycin 2 µg, and chloramphenicol 30 µg were applied (Liofilchem, Italy). For *K. pneumoniae* and *Serratia liquefaciens* strains, amoxicillin–clavulanic acid 20/10 µg, piperacillin–tazobactam 30/6 µg, cefotaxime 5 µg, ceftazidime 10 µg, ertapenem 10 µg, imipenem 10 µg, meropenem 10 µg, ciprofloxacin 5 µg, gentamicin 10 µg, and trimethoprim/sulfamethoxazole 1.25/23.75 µg were applied (Liofilchem, Italy). For *Acinetobacter* spp., piperacillin–tazobactam 30/6 µg, ceftazidime 10 µg, imipenem 10 µg, meropenem 10 µg, ciprofloxacin 5 µg, and amikacin 30 µg were applied (Liofilchem, Italy). For *Acinetobacter* spp., imipenem 10 µg, amikacin 30 µg, gentamicin 10 µg, trimethoprim/sulfamethoxazole 1.25/23.75 µg, ciprofloxacin 5 µg, and levofloxacin 5 µg were applied (Liofilchem, Italy). The size of the zone of inhibition around the disk was measured after 16–20 h of incubation. The evaluation of the results was carried out according to the European Committee on Antimicrobial Susceptibility Testing (EUCAST) standard, actual EUCAST version [17]. 

### 2.4. Biofilm Growth Using Crystal Violet Assay

Isolated Gram-positive strains were suspended in trypticase soy broth (TSB) supplemented with additional 1% glucose, and Gram-negative strains were suspended in Luria–Bertani (LB) broth for incubation at 37 °C for 16–18 h. Inoculated broths were diluted with sterile TSB or LB broths at a ratio of 1:100. Then, 150 µL measures of the diluted suspensions were transferred with a multichannel pipette into sterile 96-well plates (Thermo Scientific™ Nunc MicroWell 96-Well Microplates, flat bottom, Thermo Fisher Scientific, Roskilde, Denmark). Each plate contained 11 strains, and the negative control (uninoculated broth) contained 8 wells per strain; each experiment was performed in triplicate. The inoculated plates were cultivated aerobically at 37 °C for 48 h. After incubation, all wells were emptied by gently throwing out the liquid into a clinical waste bag without the use of a pipette. Each well was rinsed three times with sterile 250 µL 0.9% saline. After washing, staining was performed by adding 150 µL of 0.1% crystal violet per well. After 15 min, the color was removed by gently throwing out the color, and each well was washed three times with 250 µL distilled water. Finally, 150 µL of 96% ethanol was added to each well. Afterwards, the optical densities (ODs) of the wells were measured at a wavelength of 570 nm using a microplate spectrophotometer (Tecan Infinite F50, Mannedorf, Switzerland, with Magellan™ reader control and data analysis software V 6.6) [18].

### 2.5. Biofilm Calculation

The OD values for each strain were averaged and expressed as numbers. The cut-off value (ODc) was defined as three standard deviations above the mean OD of the negative control and was separately calculated for each plate. Strains were divided as follows: OD ≤ ODc = biofilm nonproducer, ODc < OD ≤ 2 × ODc = weak biofilm producer, 2 × ODc < OD ≤ 4 × ODc = moderate biofilm producer, and 4 × ODc < OD = strong biofilm producer [19].

### 2.6. Statistical Analysis

Statistical analyses were performed using IBM SPSS Statistics version 26 (Chicago, IL, USA) and Microsoft Excel 10 (Microsoft, Redmond, WA, USA). For all of the hypotheses tested, a *p*-value of less than 0.05 indicated statistical significance.

## 3. Results

### 3.1. Patient Data

The study group included 52 females (57%) and 39 males (43%) aged between 19 and 29 years (mean, 21.2 ± 1.41 years, median, 21 years).

### 3.2. Diversity of Isolated Microorganisms

Of the 91 participant samples examined, a positive cultivation finding (at least one pathogen or potential pathogen) was detected in 54 participant samples (59.3%) (Table 1). The cultivation finding was negative, i.e., only common oropharyngeal microbiota were cultivated, in 37 participant samples (40.7%) (Table 1). The most commonly isolated pathogenic bacterium was *S. aureus*, which was isolated as the only microorganism or co-isolated with other potentially pathogenic microorganisms in 41 participant samples (45%) (Table 1). Gram-positive bacteria were predominant, but at least one Gram-negative bacterium was detected in 16 samples (17.6%). Among the Gram-negative bacteria, *K. pneumoniae* was the most common, and it was isolated in seven samples.

### 3.3. Biofilm Growth and Associations

Forty-one (41) *S. aureus* strains and fifteen strains of Gram-negative bacteria were tested for biofilm production. *S. aureus* strains were predominantly biofilm producers: 25 out of 41 (61%) *S. aureus* strains were moderate or strong biofilm producers, and 14 out of 41 (34.1%) *S. aureus* strains were weak biofilm producers, but 2 out of 41 (4.9%) *S. aureus* strains were biofilm nonproducers (Figure 1). Among the Gram-negative bacteria, 6 out of 15 (40%) strains were moderate or strong biofilm producers and 6 out of 15 (40%) strains were weak biofilm producers, but 3 out of 15 (20%) strains were biofilm nonproducers (Figure 2).

A summary of the study participants’ microbiological data is shown in Table 2. There was a statistically significant association found between the presence of Gram-positive bacteria and a biofilm-formation phenotype. If a Gram-positive microbe was present, there would most likely be a biofilm-formation phenotype (Pearson χ2 test, *p* < 0.001) (Table 2).

### 3.4. Antibacterial Susceptibility

The tested *S. aureus*, *K. pneumoniae*, *P. aeruginosa*, and *Acinetobacter* spp. strains were sensitive to commonly used antibiotics, for example, amoxicillin–clavulanic acid, clindamycin, or ciprofloxacin (Table 3 and Table 4). Gram-negative rods were sensitive to all antibiotics tested (Table 4). Only one *Acinetobacter junii* strain was resistant to amikacin. One of the isolated *S. aureus* strains was methicillin-resistant *S. aureus* (MRSA), which was resistant to cefoxitin. It was a strong biofilm producer. None of the isolated *K. pneumoniae* strains were extended-spectrum beta-lactamase (ESBL) producers. No statistically significant correlations were noted between the antibiotic susceptibility pattern and the biofilm production capacity.

## 4. Discussion

The oropharynx provides heterogeneous niches for bacterial colonization. Since tonsillar infection may stem from bacteria within tonsillar crypts or the parenchyma rather than from those on the surface, we focused on the microbiota in tonsillar crypts as the most critical region for the development of tonsillopathies [1,6]. In our study of healthy individuals, we isolated and analyzed such non-oral and respiratory pathogens as *S. aureus*, *K. pneumoniae*, *P. aeruginosa*, and *Acinetobacter* spp. [9,10].

The primary ecological niche for *Staphylococcus* is the nostrils; nevertheless, the oral cavity comprises a significant reservoir for these bacteria, and some adults exhibit exclusive oral colonization [20]. In a study by Albrich and Harbarth, the colonization of extranasal sites was associated with the persistent carriage of *S. aureus* [21]. In a study carried out in the USA by Hanson and colleagues, it was reported that 6.2% of adults carried *S. aureus* only in the anterior nares, 18.6% only in the oropharynx, and 19.8% in both sites [22]. A carrier rate in the oral cavity from 17 to 48% has been reported within student populations [23,24]. Healthy Swedish dental students had an *S. aureus* prevalence of 44.6%; no MRSA was detected among them [24]. Healthcare workers were found to carry MRSA at a rate of 23.7% [21]. The prevalence of MRSA in healthy carriers has been reported to range from 1.5 to 26% [25,26,27]. In our study, *S. aureus* was the most common pathogen isolated; it was isolated in 45% of cases, and MRSA was isolated in 1.1% of cases, which is in accordance with previous studies.

In a study by Jeong and colleagues, *K. pneumoniae* was isolated from the tonsillar core samples of recurrent tonsillitis patients in 6.7% of cases, and from those of tonsillar hypertrophy patients in 1.5% of cases [28]. Our study showed that *K. pneumoniae* was present in the tonsillar crypt specimens of healthy subjects in 7.7% of cases.

Several studies have analyzed the role of extracellular or intracellular *P. aeruginosa* in the origins of periodontal or pulmonary diseases [29,30]. In one study, *P. aeruginosa* was the third most common pathogen after *E. faecalis* and *S. aureus* in human buccal and gingival epithelial cells obtained from subjects with periodontitis and periodontally healthy subjects; no difference was observed in the prevalence of *P. aeruginosa* between periodontitis and periodontally healthy subjects or between the types of epithelial cells [31]. In another, *P. aeruginosa* was detected at high mean prevalence and counts in the subgingival microbiota and was closely related to periodontal inflammation and tissue destruction [32]. Other studies have reported a 1.4–3.8% prevalence of *P. aeruginosa* in the tonsillar samples of recurrent tonsillitis patients [28,33,34], and a 0.9% prevalence in the tonsillar samples of tonsillar hypertrophy patients [28]. Our study showed that *P. aeruginosa* was present in the tonsillar crypt specimens of healthy subjects in 2.2% of cases. Among *Acinetobacter* strains, *Acinetobacter baumannii* was not detected. The variety and prevalence of pathogen isolation from tonsil samples may vary depending on the sampling method; for example, tonsil surface swabs may be less informative than tonsil crypt material [35].

Tonsillar crypts are a suitable site for biofilm formation. Tonsillar crypts are able to collect debris, and the mineralization of this debris leads to tonsillolith formation [36]. Tonsilloliths possess dynamic biofilms similar to dental biofilms [37]. Our study showed that in healthy subjects, 61% of *S. aureus* strains and 40% of Gram-negative bacteria strains were moderate or strong biofilm producers. Our study confirms that biofilm formation is a normal bacterial lifestyle, and that biofilms can exist in the tonsils of healthy individuals. In the study by Penesyan and colleagues, biofilm was described as a main microbial lifestyle; biofilms perform an important function for microbes by providing a protective environment in which genotypic and phenotypic diversity is generated before being released [38]. Biofilm characteristics may differ between diseased and healthy individuals. Chervinets and colleagues reported that the microbiota of the oral cavities of patients with periodontitis had a greater ability to adhere to the cells of the mucous membrane than those of healthy people, while their ability to form biofilms and exhibit pathogenic properties was enhanced [39].

The localization of the causative agents in biofilms may contribute to antibiotic resistance. Antibiotic resistance is a major concern regarding *S. aureus*, especially MRSA. An increasing prevalence of MRSA in healthy carriers has been reported, amounting up to 21% in the nasal samples of dental students [25]. The prevalence of MRSA in the oral cavity is less known; subgingival sites and tongue surfaces were tested in a previous study, and no MRSA was detected [40]. Our study revealed one (1.1%) MRSA isolate from tonsil specimens. An important finding in this study was the high rate of benzylpenicillin- and ampicillin-resistant *S. aureus* strains isolated from healthy individuals; however, no isolates were resistant to clindamycin. The data that were obtained about *S. aureus* antibacterial resistance are in accordance with the study by Katkowska et al. [41]. Clindamycin is widely used in dentistry, and many clinics have substituted it for common penicillins (oxacillin and methicillin); clindamycin is prescribed in the case of allergy to beta-lactams [24].

It has been hypothesized that infectious strains have different virulence arsenals than those colonizing healthy individuals [24]. However, some studies have failed to show, for example, that *S. aureus* strains isolated from oral infections and noninfected controls represent different subgroups of phenotypic and genotypic characteristics [24]. It has therefore been suggested that classical opportunistic infections develop due to an imbalance within the host–parasite relationship, and that the infectious disease persists as long as the compromised condition prevails [24].

We would like to highlight certain strengths of the present study. We used tonsil brushes as an alternative, noninvasive method for collecting tonsil specimens, eliminating the need to traumatize tonsils in order to collect tonsil tissue. Therefore, in our study, we included healthy individuals without any signs of palatine tonsil disease. The limitations of this study were the absence of a comparison group, a small number of Gram-negative bacterial strains analyzed, and in vitro biofilm formation. The environmental factors (for example, temperature, pH, glucose level, type of media) influence bacterial biofilm production. We created the best possible conditions for bacterial growth and biofilm formation in vitro as described by Stepanović et al. [19]. However, complex in vivo models for biofilm studies are superior and encouraged. In further studies, it is recommended to analyze bacterial genetic factors as well, as they also play an important role in biofilm formation and would be useful for such an analysis.

## 5. Conclusions

Biofilm is a commonly observed feature that seems to be a naturally existing form of pathogenic bacteria colonizing human tissue. *S. aureus*, *K. pneumoniae*, *P. aeruginosa*, and *Acinetobacter* spp. occasionally occur in the tonsillar crypts of healthy individuals, and, therefore, it is most likely that *S. aureus*, *K. pneumoniae*, *P. aeruginosa*, and *Acinetobacter* spp. in opportunistic tonsillar infections originate from the tonsillar crypt microbiota.

## Figures and Tables

**Figure 1 microorganisms-11-00258-f001:**
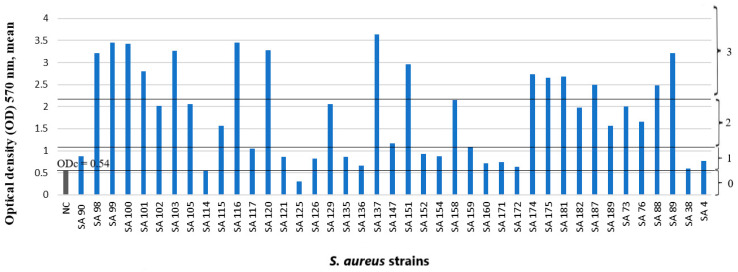
Biofilm production capability on microtiter plate of 41 *S. aureus* strains. Bars represent mean values of OD (measured at wavelength of 570 nm). Trypticase soy broth with 1% glucose as a negative control (NC). The number designates the participant; the letters indicate the strain isolated: SA—*Staphylococcus aureus*. The cut-off value (ODc) and biofilm production capacity levels are marked with horizontal lines: 0—biofilm nonproducers; 1—weak biofilm producers; 2—moderate biofilm producers; 3—strong biofilm producers.

**Figure 2 microorganisms-11-00258-f002:**
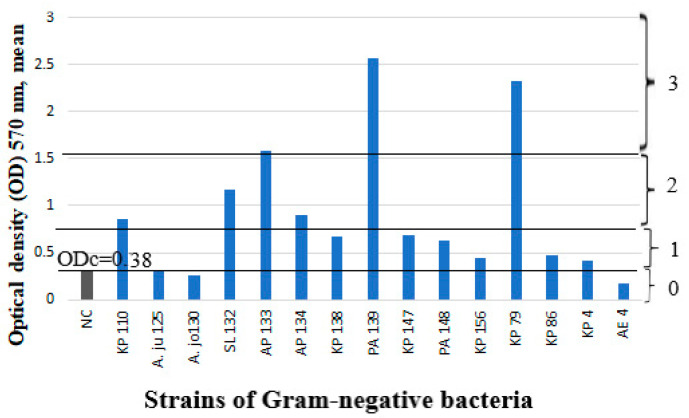
Biofilm production capability on microtiter plates of 15 strains of Gram-negative bacteria. Bars represent mean values of OD (measured at wavelength of 570 nm). Luria–Bertani medium as a negative control (NC). The number designates the participant; the letters indicate the strain isolated: KP—*Klebsiella pneumoniae*; A.ju—*A.junii*; A.jo—*Acinetobacter johnsoni*; SL—*Serratia liquefaciens*; AP—*Acinetobacter pitti*; PA—*Pseudomonas aeruginosa*; and AE—*Acinetobacter ewoffi*. The cut-off value (ODc) and biofilm production capacity levels are marked with horizontal lines: 0—biofilm nonproducers; 1—weak biofilm producers; 2—moderate biofilm producers; 3—strong biofilm producers.

**Table 1 microorganisms-11-00258-t001:** Microorganisms isolated from tonsillar crypts of 91 healthy individuals.

Combinations of Isolated Strains	Count (n)
Normal oral microbiota	37
*S. aureus* + normal oral microbiota	20
*S. aureus*	16
*S. aureus* + *Acinetobacter junii*	2
*S. aureus* + *K. pneumoniae*	1
*S. aureus* + *Candida* spp. + *Streptococcus viridans*	1
*S. aureus* + *K. pneumoniae* + *Serratia liquefaciens* + normal oral microbiota	1
*K. pneumoniae*	5
*P. aeruginosa*	2
*Acinetobacter pittii*	2
*Acinetobacter johnsonii*	1
*Serratia liquefaciens*	1
*Streptococcus dysgalactiae*	1
*Acinetobacter ewofii* + normal oral microbiota	1

**Table 2 microorganisms-11-00258-t002:** Summary of microbiological data of study participants.

Participants’ Microbiological Data		Results	*p*-Values
Isolation rate	Normal oral microbiota only, n (%)	37/91 (40.7%)	
	Gram-positive strains, n	43	
	Gram-negative strains, n	17	
Biofilms, mean OD	*S. aureus* biofilms, mean OD	1.89	
	Gram-negative microbe, mean OD	0.95	
Biofilm-producing strains	Biofilm-producing strains, n	51	
	*S. aureus* biofilm-producing strains, n	39	
	Gram-negative microbe biofilm-producing strains, n	12	
	Strong and moderate biofilm producers, n	31	
Associations between variables	Gram-positive microbe and biofilm-producing strain		*p* < 0.001
	Gram-negative microbe and biofilm-producing strain		*p* = 0.808

**Table 3 microorganisms-11-00258-t003:** Antibiotic resistance among *S. aureus* strains isolated from healthy subjects.

		Antibiotic Resistance (%)										
	Strains (n)	FOX	CRO	P	AMP	AMS	AUG	NOR	AK	E	CD	C
*S. aureus*	41	2.4	2.4	75.6	75.6	2.4	2.4	2.4	2.4	14.6	0	4.9

FOX, cefoxitin; CRO, ceftriaxone; P, benzylpenicillin; AMP, ampicillin; AMS, ampicillin–sulbactam; AUG, amoxicillin–clavulanic acid; NOR, norfloxacin; AK, amikacin; E, erythromycin; CD, clindamycin; C, chloramphenicol.

**Table 4 microorganisms-11-00258-t004:** Antibiotic resistance among *K. pneumoniae*, *P. aeruginosa*, *Serratia liquefaciens*, and *Acinetobacter* spp. strains isolated from healthy subjects.

		Antibiotic Resistance (%)											
	Strains (n)	AUG	TZP	CTX	CAZ	ETP	IMP	MEM	CIP	GM	SXT	AK	LEV
*K. pneumoniae*	7	0	0	0	0	0	0	0	0	0	0		
*P. aeruginosa*	2		0		0		0	0	0			0	
*Acinetobacter* spp.	5						0		0	0	0	20	0
*Serratia liquefaciens*	1	0	0	0	0	0	0	0	0	0	0		

AUG, Amoxicillin–clavulanic acid; TZP, Piperacillin—tazobactam; CTX, Cefotaxime; CAZ, Ceftazidime; ETP, Ertapenem; IMP, Imipenem; MEM, Meropenem; CIP, Ciprofloxacin; GM, Gentamicin; SXT, Trimethoprim-sulfamethoxazole; AK, Amikacin; LEV, Levofloxacin.

## Data Availability

The datasets generated are available from the corresponding author upon reasonable request.

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
