# Peer review of "In Vitro Assessment of Biofilm Production, Antibacterial Resistance of Staphylococcus aureus, Klebsiella pneumoniae, Pseudomonas aeruginosa, and Acinetobacter spp. Obtained from Tonsillar Crypts of Healthy Adults"

_microorganisms, 2023, doi:10.3390/microorganisms11020258_

Round 1

Reviewer 1 Report

The authors wrote a manuscript detailing a study on the characterization of bacteria in tonsillar crypts.  The authors are encoraged to address the following comments:

1. Title - Is misleading. The title suggests that the biofilms were formed by bacteria in tonsillar crypts. In actuality, the paper focuses on in vitro generation of biofilms derived from cultures obtained from tonsillar crypts. The title should be modified.

2. Experimental design - why was culture used instead of PCR?

3. Why did the authors choose not to disclose the other bacteria present in the tonsillar crypt and focus only on specific bacteria? What percentage of the tonsillar crypt microbiome were the organisms studied as opposed to total bacteria content?

4. What was the significance of studying the biofilms created in a laboratory setting? How does the laboratory setting inform that which happens in a patient?

5. How do laboratory strains compare to the strains of bacteria isolated in terms of antibiotic susceptibility?

6. How do laboratory strains compare to clinical strains in terms of biofilm formation?

7. The authors should address the key significance of the study.

Reviewer 2 Report

The manuscript has many essential failures concerning antibiotic susceptibility testing of isolated bacteria from tonsillar crypts.

1.      First, the description of antibiotic susceptibility determination is missing from "Materials and Methods". Why?

2.      Cefoxitin is not applicable for the treatment of staphylococcal infections, it is only used to detect resistance to methicillin with high sensitivity and specificity. Therefore, cefoxitin should not be mentioned as the Authors did in the Abstract line 27, or the Results, line 196. 

3.      It is also obvious that MRSA strains are by definition resistant to all β-lactam antibiotics (cefoxitin, ceftriaxone, benzylpenicillin, ampicillin, ampicillin–sulbactam, and amoxicillin with clavulanic acid), such as those listed in lines 196-198.

4.      It is difficult to believe that S. aureus strains susceptible to all antibiotics were intermediate-susceptible (de facto resistant) to ciprofloxacin, it can be assumed that it is a misinterpretation of the results of antibiograms. It can be supposed that the Authors used the wrong guidelines, which they did not mention at all.

5.      It is unnecessary to present the drug susceptibility of each isolated bacterial strain as in Table 3 and Table 4, they do not provide anything important or interesting, there should be created Tables with the percentages of drug-resistant strains.

6.      The Authors in Results section also omit the description and interpretation of antibiotic susceptibility of the isolated Gram-negative rods.

7.      The Authors studied the susceptibility to antibiotics to which the Gram-negative rods are naturally resistant. The study should be repeated by current EUCAST and/or CLSI guidelines.

8.      The Authors did not detect any mechanisms of resistance in Gram-negative rods except ESBL, such as KPC, and MBL. Why? This should be studied.

The manuscript cited the reference about the composition of the microflora in cancer (SCC) and did not provide the relevant current literature on S. aureus, of which there were most,  https://pubmed.ncbi.nlm.nih.gov/27862322/

The conclusions drawn in the manuscript can be given without performing the studies and based on previous literature.

Reviewer 3 Report

This article presents an evaluation of the biofilm formation, antibacterial resistance of S. aureus, K. pneumoniae, P.aeruginosa and Acinetobacter spp. in Tonsillar Crypts of healthy adults.

Major concern:

·       the scientific value of this work is low. It has no novelty and some parts are very confused. The authors lacks of basic microbiological knowledge regarding antimicrobial susceptibility of these pathogens. The chosen methods are not the standard in daily routinely labs and do not offer the information needed. The reader will have not benefit whatsoever reading this manuscript. Contrary, he/she will be very confused.

·       Line 18-21 – there is no hypothesis formulated and the aim of the study is very vague.

·       Line 25-28 – it cannot possible be that K. pneumoniae, Acinetobacter and Pseudomonas are sensitive to clindamycin. On the other hand, the latter two cannot be sensitive to amoxicillin-clavulanic acid.

·       The conclusion in abstract is very different from the conclusion of the manuscript!

·       There is difficult to judge based on the method section if the methods were correctly chosen.

·       No definition of potential pathogen was given

·       How it is possible that no oral microbiota was found in so many samples?

·       Table 1 presents oral microbiota to be found in 58% of the samples but table 2 only in 40,7% of the samples. How do the authors explain that?

·       The results are misleading. The authors presents constitutive resistance as acquired resistance!?

·       The authors presents as a result that Pseudomonas and Acinetobacter are no producing ESBL. They are never producing ESBL. They should have not be tested of ESBL production.

·        

Minor concern

·       Line 84-87 from introduction belong to method section

·       The method section contains some strange word like “interoperated” , better performed or expression like “each plate consisted of 11 strains”, better each plate contain 11 strains

·       There is no intermediary resistant strains (see EUCAST new definition from 2019)

Round 2

Reviewer 2 Report

Accept in the present form

Reviewer 3 Report

Accept in the present form.